# Validation of the European Obstructive Sleep Apnea Screening (EUROSAS) in Professional Male Drivers

**DOI:** 10.3390/jcm13195976

**Published:** 2024-10-08

**Authors:** Yeliz Celik, Semih Arbatli, Baran Balcan, Yuksel Peker

**Affiliations:** 1Koc University Research Center for Translational Medicine (KUTTAM), Koc University, Istanbul 34010, Turkey; yecelik@ku.edu.tr; 2Graduate School of Health Sciences, Koc University, Istanbul 34450, Turkey; sarbatli20@ku.edu.tr; 3Department of Pulmonary Medicine, Koc University School of Medicine, Istanbul 34450, Turkey; mbalcan@kuh.ku.edu.tr; 4Division of Pulmonary, Allergy, and Critical Care Medicine, University of Pittsburgh School of Medicine, Pittsburgh, PA 15213, USA; 5Department of Clinical Sciences, Respiratory Medicine and Allergology, Faculty of Medicine, Lund University, 22185 Lund, Sweden; 6Department of Molecular and Clinical Medicine, Institute of Medicine, Sahlgrenska Academy, University of Gothenburg, 40530 Gothenburg, Sweden

**Keywords:** OSA, drivers, motor vehicle accident, EUROSAS

## Abstract

(1) **Background:** The European Union Driver License Committee recently developed a questionnaire as a screening tool for obstructive sleep apnea (OSA), named the European Obstructive Sleep Apnea Screening (EUROSAS) questionnaire for drivers. The aim of the current study was to investigate the diagnostic performance of the EUROSAS to predict risk of OSA in professional male drivers. (2) **Methods:** Fifty-eight drivers were included in the current study. All participants were asked to fill out the EUROSAS before an overnight polysomnography (PSG) in the hospital. OSA was defined as an apnea-hypopnea index (AHI) 5 events/hour on the PSG. (3) **Results:** Out of 58 participants, the EUROSAS correctly identified 39 (67.2%) cases as having high-risk OSA and one patient as having low-risk OSA, using AHI ≥ 5 events/h. The results indicated that the EUROSAS has a sensitivity of 67.2%, a specificity of 33.3%, a positive predictive value of 94.8%, and a negative predictive value of 5.2%. Similar results were obtained using AHI cut-offs of 15 and 30 events/h. (4) **Conclusions:** The EUROSAS provides a moderate level of accuracy for the screening of OSA in the professional male drivers. It seems that the diagnostic performance of the EUROSAS is not promising as an alternative questionnaire to identify professional drivers with OSA, probably due to participant response bias. Despite its limited evidence, the EUROSAS might have potential as a clinical screening tool in the general population.

## 1. Introduction

Motor vehicle accidents (MVAs) are one of the most important causes of fatal injuries and death worldwide [1]. One of the significant risk factors contributing to MVAs is obstructive sleep apnea (OSA), which is a health-related condition characterized by repetitive episodes of airflow cessation due to upper airway collapse during sleep [1]. The estimated prevalence of OSA ranges from 9% to 38%, while 80–90% of those individuals with OSA are frequently undiagnosed in the general population [2,3]. Given the consequences of OSA, such as impairment of executive functions, attention, psychomotor speed, and coordination, it is not surprising that the risk of MVAs in OSA population is greater than that for the general population.

OSA appears to be much more prevalent in professional motor vehicle drivers due to unique population characteristics, such as a higher proportion of males, obesity rates, and age distribution [4]. Indeed, the recent literature has demonstrated the association of those characteristics with low arousal threshold as an underlying mechanism of OSA, which is defined as an increased propensity to be awakened prematurely in response to breathing disturbances [5,6]. These characteristics may also explain the high occurrence of OSA in commercial drivers. The incidence rate of OSA has been reported to be between 28% and 78% in commercial drivers [7]. A large number of studies have reported that commercial drivers are considered to be a population at high risk of OSA, with a two- to five-fold greater risk of traffic accidents and a three to five times stronger chance of personal injury [8,9,10,11,12]. Furthermore, the drivers with untreated OSA are three to seven times more likely to be involved in occupational accidents in addition to MVAs [13,14,15]. Thus, assessment of the OSA risk and development of effective methods to identify and treat commercial drivers with OSA is essential to reduce MVAs.

Considering the significant public health and safety burden of OSA, early identification of the professional drivers with unrecognized OSA seems to be crucial. An overnight hospital polysomnography (PSG) is the gold standard to determine the presence and severity of OSA. Given the high cost and poor availability of PSG as well as long waiting lists for PSG in hospitals, the systematic evaluation of professional drivers using PSG is not feasible in clinical settings. The use of self-reported questionnaires as a first step in diagnosing OSA might be an option to manage this issue.

Several questionnaires have been developed to identify OSA in the general population and clinical cohorts, including the Berlin Questionnaire (BQ), the STOP-Bang Questionnaire (SBQ), and the STOP Questionnaire (SQ) [16]. Though these questionnaires were used for professional drivers, none of them was considered specific for drivers in the occupational setting. In this context, the European Union Driver License Committee developed a questionnaire as a screening tool for OSA, especially for drivers of motor vehicles (Appendix A) [17]. Previously, we named the questionnaire the European Obstructive Sleep Apnea Screening or “EUROSAS”, and reported that its test–retest reliability was poor among male and female drivers [18]. Further validation study was needed due to lack of objective sleep testing [18]. Therefore, the aim of the current study was to validate the EUROSAS questionnaire as a screening tool to identify those drivers with a risk of OSA and an urgent need for objective PSG testing. A potential validated tool would improve the accuracy of OSA diagnosis and treatment among professional drivers and potentially decrease the number of MVAs worldwide.

## 2. Materials and Methods

### 2.1. Study Design and Participants

The current research compromises 58 professional male drivers included in a study on driving simulation conducted in the Koç University Hospital Sleep Laboratory, Istanbul. Each participant provided written informed consent and was asked to fill out questionnaires before undergoing an overnight hospital PSG. The inclusion criteria were holding a driving license longer than 3 years and having been driving actively for at least 6 or 7 days per week. Participants having acute illness and no longer holding a valid driver’s license were deemed ineligible. Each subject was invited to voluntarily participate in the current research. The detailed information regarding the simulator test has been described elsewhere [19,20]. The study protocol was approved by the Committee on Human Research (2020. 292.IRB2.083; 19 June 2020).

### 2.2. Data Collection and Definition

Demographics and baseline characteristics of the participants as well as comorbidities were reported. Obesity was classified in 3 categories based on the latest recommendation of the World Health Organization [21]. Class I obesity was defined as a BMI of 30 to less than 35 kg/m^2^, Class II obesity as BMI 35 to less than 40 kg/m^2^, and Class III obesity as BMI 40+ kg/m^2^. All participants filled out questionnaires regarding sleep-related symptoms and sleep habits, which were used in clinical routines. The alcohol consumption criteria were using alcohol “every day” or “once a week”.

### 2.3. EUROSAS

The EUROSAS is a self-rating Likert scale including 11 items, used to identify individuals with a high risk of OSA [17]. The questionnaire consists of 11 items with 3 options to reply (yes/no/don’t know). The first four items correspond to demographic characteristics of the responder, including age (item 1), gender (item 2), and BMI (item 3 and 4), while the following six items refer to history of sleepiness while driving (item 5), past MVAs due to EDS (item 6), symptoms of snoring (item 7), witnessed apneas (item 8), non-restorative nocturnal sleep (item 9), and hypertension (item 10). The last item corresponds to a total score of the ESS, which is a widely used questionnaire to assess excessive daytime sleepiness in sleep clinic populations. Values were attributed to each option, reflecting the strength of the relationship between a given answer and the risk for MVAs or the probability of suffering from OSA, based on a consensus minimal agreement among the members of the working group. The possible value for the EUROSAS ranges from 2 to 24, and individuals with 10 points or above on the EUROSAS scale were considered to be positive for OSA [17].

### 2.4. Sleep Measurement

A full night PSG (NOX-A1 system; Nox Medical Inc., Reykjavik, Iceland) was used to record sleep measurements in the present study. The PSG testing includes EEG, EOG, chin and leg electromyograms, nasal airflow, heart rate, oxyhemoglobin saturation (SpO_2_), snoring intensity, body position, and thoraco-abdominal and leg movements. A oronasal thermal sensor was used to detect temperature changes in breath to monitor airflow. A nasal pressure transducer was included to detect pressure changes in breath to monitor airflow. Two respiratory inductance plethysmography (RIP) belts were used to observe respiratory effort in the thorax and abdomen. Oxygen saturation was obtained through pulse oximetry. A single electrocardiogram was used to monitor heart rate and rhythm. Leg movements were detected via electrodes placed on the left and the right anterior tibialis muscles. Body position was documented in the lab by the sleep technician using video confirmation or by using a position monitor attached to the patient. As recommended in the AASM Manual for the Scoring of Sleep and Associated Events, arousals and sleep stages were scored using 30 s epochs [22]. Considering the latest recommendations of the AASM, apnea was defined as an almost complete (≥90%) cessation of airflow, while hypopnea was defined as a decrease in nasal pressure amplitude of ≥30% and/or thoraco-abdominal movement ≥30% for ≥10 s if there was a significant oxyhemoglobin desaturation (reduction by ≥3% from the immediately preceding baseline value) and/or an arousal [22]. The average and minimum SpO_2_ values as well as the time spent below 90% SpO_2_ were also reported. The oxygen desaturation index (ODI) was calculated as the number of significant desaturations per hour of total sleep time. All PSG recordings were manually scored in a mixed order by a certified sleep technician under supervision of YP, blinded to the EUROSAS categorizations.

### 2.5. Statistical Analysis

Baseline characteristics of the participants and the PSG measurements were summarized using a median with 25th and 75th percentiles for the continuous variables, and as a count with percentage for the categorical variables. The Shapiro–Wilk test was used to test normality assumptions. The differences between participants with OSA and without OSA were compared using the Mann–Whitney rank sum test for continuous variables, and χ^2^ test or Fisher’s exact test for categorical variables. The EUROSAS was validated against the PSG measurements. Based on the International Classification of Sleep Disorders—3, OSA was defined using 3 different thresholds, including AHI ≥ 5 events/h, ≥15 events/h, and ≥30 events/h [23]. Using those thresholds, the diagnostic performance of the EUROSAS was evaluated regarding the diagnostic odds ratio, disease prevalence, sensitivity, specificity, negative likelihood ratio, negative predictive value, positive likelihood ratio, positive predictive value, and accuracy. Area under receiver-operating characteristic (ROC) curve analysis was conducted to examine the association between continuous AHI values measured with the PSG and the EUROSAS results (low and high risk), as well as to predict the best AHI cut-off value. The internal validity was examined by means of Cronbach’s α value of the EUROSAS’s 11 items. The accepted significance level was 5%, and statistical analyses were performed using IBM SPSS 28.0 for Windows SPSS Inc., Chicago, IL, USA.

## 3. Results

### 3.1. Baseline Characteristics of the Study Population

Baseline characteristics of the study population as well as the study groups are presented in Table 1. A total of 58 male professional drivers with a mean age of 46.9 (7.5) were included in the present study. The median ESS score was 6.5 (IQR 3.0–10.3), and only 24.1% of the cohort was classified as sleepy (ESS score ≥ 11). The participants with OSA were more obese than the ones without OSA, while other demographic characteristics and comorbidities were similar.

### 3.2. Distribution of the Participants’ Responses on the EUROSAS

Distribution of the participants’ responses on the EUROSAS is presented in Figure 1A. The highest mean score was yielded on item 1, since the current sample consisted of male drivers. Apart from this result, the mean score of the item regarding BMI was rated as the highest compared to the rest of the scale. The lowest mean score was calculated for item 6, “Did you have a serious accident due to sleepiness in the last 3 years?”. As illustrated in Figure 1B, the proportion of negative answers scored as positive was high on item 9, “Do you usually wake up refreshed after a full night sleep?”.

The standardized Cronbach’s alpha value was 0.176, indicating poor reliability of the questionnaire. Removing any of those items from the scale did not increase the Cronbach’s alpha value, since each item contributed poorly to the overall reliability of the EUROSAS, being all below 0.36.

### 3.3. Categorization of the Participants with High Risk of OSA

Out of 58 participants, 39 (67.2%) cases were categorized as having a high risk of OSA based on the EUROSAS, while 19 (32.8%) were classified as having a low risk of OSA. As presented in Table 2, the baseline sleep characteristics of drivers in the high risk of OSA group were similar to those with low risk of OSA.

Figure 2 shows that the total EUROSAS scores in the patients with OSA and no OSA were similar. There was a positive poor correlation between the total EUROSAS scores and the AHI events/h (Pearson Correlation: 0.217, *p*: 0.101).

### 3.4. Performance of the EUROSAS in Detecting OSA

As illustrated in Figure 3, 37 patients (94.9%) in the high-risk OSA group and 18 patients (94.7%) in the low-risk OSA group had OSA using the AHI cut-off of 5 events/h on the PSG. The corresponding values were 26 (66.7%) vs. 13 (68.4%) for the AHI cut-off of 15 events/h, and 17 (43.6%) vs. 8 (42.1%) for an AHI cut-off of 30 events/h.

Cohen’s Kappa values as a measure of agreement were calculated as 0.002 (*p* = 0.98) for the AHI cut-off of 5 events/h, −0.018 (*p* = 0.89) for the AHI cut-off of 15 events/h, and 0.012 (*p* = 0.92) for the AHI cut-off of 30 events/h, which indicates poor agreement between the EUROSAS categorization and OSA classification using the AHI as a cutoff.

As seen in Table 3, the predictive values of the EUROSAS were calculated using different AHI thresholds. The diagnostic accuracy of the EUROSAS was the highest using the AHI cut-off of 5 events/h. At this threshold, the EUROSAS had a sensitivity of 67.2%, a specificity of 33.3%, a positive predictive value of 94.8%, and a negative predictive value of 5.2%. Corresponding values at the AHI thresholds of 15 and 30 were slightly similar. On the other hand, the diagnostic odds ratio was the highest when the AHI cut-off of 30 events/h was applied (Table 3).

Figure 4 illustrates the ROC curve of the association between the continuous AHI values and the EUROSAS results (high- vs. low-risk OSA). The area under the curve was 0.46 (95% CI 0.29–0.62), indicating a mediocre performance of the EUROSAS. The highest sensitivity and specificity rate were obtained at the AHI cut-off of 14.6 events/h based on the ROC curve analysis.

## 4. Discussion

The main finding of the current study is that the EUROSAS has a moderate predictive ability to detect OSA in professional male drivers. The sensitivity and specificity rates were similar under the different AHI thresholds, whereas the best performance of the positive and negative predictive values was obtained using the AHI cut-off of 5 events/h. The EUROSAS correctly identified 67% of the patients who have mild to severe OSA, as well as 33% of the participants who had no OSA. As reflected by the Cronbach’s alpha coefficients, the reliability of the EUROSAS questionnaire was poor in the current population (α = 0.18). In line with those results, the ROC analysis revealed that EUROSAS was not reliable as a screening tool for OSA in professional male drivers.

To the best of our knowledge, this is the first validation study of the EUROSAS using the PSG to diagnose OSA in professional motor vehicle drivers. Recently, the diagnostic performance of the EUROSAS as well as the available screening tools for OSA, including the BQ, STOP-Bang, and ESS, were compared against the respiratory event index (REI) value measured by Home Sleep Apnea Testing (HSAT) in a cohort of male commercial drivers [24]. The main finding of this study was that none of the OSA questionnaires, including the EUROSAS, BQ, SBQ, SQ, and the ESS, are suited for assessing OSA in commercial drivers. Although the EUROSAS has a high specificity to predict risk of mild to severe OSA (REI ≥ 5) as well as moderate to severe OSA (REI ≥ 5), (93% and 87%, respectively), its diagnostic performance has been evaluated as poor regarding the values for sensitivity (19% and 28%, respectively) [24]. The differences between that study and the current findings might be mainly attributable to the use of different overnight sleep recordings (PSG vs. HSAT), different reference standards (REI vs. AHI), sample size differences (*n* = 315 vs. *n* = 58), and participant characteristics such as incidence of obesity (21.9% vs. 67%) and comorbidities.

Several studies have previously been conducted to evaluate the accuracy of the aforementioned OSA questionnaires in male drivers. A retrospective study conducted by Ueyama et al. examined 1309 commercial drivers with ESS and reported that 60% of the study population had moderate to severe OSA based on the PSG but only 9% of them had an ESS score over 10 points due to reduced awareness of subjective daytime sleepiness [25]. Fırat et al. evaluated the usefulness of four established OSA questionnaires using the AHI measured with PSG in highway bus drivers [26]. They reported that the SBQ performed better to identify bus drivers with a risk of OSA compared to the BQ and SQ questionnaires. In another study conducted by Popevic et al., SBQ had a high performance in identifying commercial drivers with a high risk of OSA [27]. Notwithstanding this, the efficacy of those screening tools is still debated due to poor symptom awareness and denial of symptoms as well as increased prevalence of OSA in professional drivers [24]. With the exception of these three studies as well as the current work, the previous reports included only participants who were categorized as having a high risk of OSA based on the questionnaires.

The SBQ is a questionnaire comprising the four subjective items of STOP (Snoring, Tiredness, Observed apnea, and high blood Pressure) and the four objective items of Bang (BMI, age, neck circumference, male gender). A score of five and above is identified as high risk of OSA. It has been primarily used to predict OSA-related postoperative risks in surgical populations but also has been widely used for OSA screening across different populations,, including patients with chronic diseases such as multiple sclerosis and kidney failure, veterans, and patients referred to sleep clinics [28]. Both the SBQ and the first part of the questionnaire (STOP) has been reported as an appropriate screening tool to determine OSA in previous meta-analyses and systematic review studies [28,29]. Although the SBQ has been validated in the general population as well as in surgical and sleep clinic patients, evaluation of its diagnostic utility in commercial drivers is limited [28]. As aforementioned, Popevic et al. conducted a study to validate a Serbian version of the SBQ in a group of commercial drivers and reported the sensitivity and specificity as 86% and 53% for an AHI ≥ 5, 100% and 40.3% for AHI ≥ 15, and 100% and 35% for AHI ≥ 30 [27]. For the positive and negative predictive values, corresponding parameters across the different AHI thresholds were 71% and 74%, 33% and 100%, and 17% and 100%, respectively. Supporting evidence was also provided by a recent bivariate meta-analysis, demonstrating that the SBQ is a more accurate screening tool for detecting OSA compared with the BQ and ESS. It was argued that the superior performance of the SBQ might be due to the content of SBQ assessing not only sleep-related symptoms as in the BQ and SB but also clinical characteristics, such as BMI, age, neck circumference, and gender, capturing the entire spectrum of OSA (mild to severe) [27]. Despite the fact that the EUROSAS consists of all items in the SBQ, it is surprising that its diagnostic accuracy is unsatisfactory.

The BQ is another self-administered questionnaire used widely to identify patients with OSA, comprising 10 items in three subcategories and assessing the presence and frequency of snoring behavior and/or witnessed apneas during sleep (Category I), the presence of daytime sleepiness (Category II), and the patients’ history of hypertension and/or obesity (Category III) [30]. In the study by Popevic et al., the BQ had a sensitivity of 51% and a specificity of 86% for an AHI ≥ 5 among commercial drivers [31]. Corresponding parameters for the positive and negative predictive values were 83% and 57%, respectively. They concluded that the BQ has good measurement properties as an OSA screening tool in commercial drivers [31].

Recently, Huhta et al. established a new screening tool named BAMSA, which was created using four widely used sleep questionnaires, including the BQ, STOP-Bang, NoSAS, and ESS [32]. The validation of the BAMSA was conducted using cardiorespiratory polygraphy rather than PSG. Although 2066 professional drivers were included in the study, only 172 of them participated in the sleep testing. Their results showed that the BAMSA is not only more sensitive (85.7%) but also more specific (78.8%) than the other questionnaires in detecting OSA in professional truck drivers [32]. The better performance of the BAMSA compared to the EUROSAS might be due to its methodology; the items included to the BAMSA were selected after examining ROC analysis of the four questionnaires and its performance rather than testing a prior hypothesis.

Compared to the SBQ, the BQ, and the BAMSA, the EUROSAS has a weaker diagnostic accuracy to predict OSA in professional drivers. Several arguments could be discussed to explain the lower performance of the EUROSAS in the present study. First, the majority of the previous studies were retrospective, including patients recruited from sleep clinics, implying that such patients may suffer from severe OSA and they might have other sleep disorders and related symptoms along with OSA. The severity of the target disease might also affect the diagnostic accuracy of its screening tool [33]. It is likely that the estimates of diagnostic accuracy are to be favorable when the study sample includes a high proportion of subjects with more severe conditions. Thus, moderate OSA severity in the current sample might have an impact on the diagnostic accuracy of the EUROSAS. Second, part of the variability might be attributable to differences in population characteristics related to age, sex, comorbidities, etc., and to the limited sample size of the current study. This is supported by recent evidence indicating a large proportion of OSA cohorts reporting their symptoms as minimal or reporting no symptoms [34]. Studies have also found no association between AHI and self-reported sleep-related symptoms. Third, given that the participation was voluntary, the participants’ responses on the items of EUROSAS were open to bias. It is likely that current drivers would underestimate their OSA severity due to legal issues. A previous study supported this argument, reporting that some commercial drivers did not report OSA symptoms in case it was not safe to do so [35]. In line with this, previous studies have indicated that professional drivers usually underreport OSA symptoms or show resistance to OSA assessment, in order to avoid potential consequences on medical certification and employment, economic implications of additional testing, and occupational consequences of lost work time [26,27,35].

Our study has several limitations. The current sample size is small and cannot be considered representative even though there is no consensus regarding the optimal sample size for validation studies. It has been reported that a subject per item ratio of 5 to 10:1 is acceptable [36]. The proportion of the participants who report smoking is relatively high, which might have an impact on the current findings regarding the confirmed association between smoking and OSA [37]. Only male participants were included; thus, the findings might not be applicable for female drivers or generalized for the general population. Additionally, the majority of the current sample were taxi drivers who were registered in the same area, which might lead to sample selection bias in the current study. Further validation studies are needed to determine the diagnostic accuracy and cost effectiveness of using the EUROSAS in the general population as well as in primary care settings.

## 5. Conclusions

The EUROSAS questionnaire has a moderate level of accuracy for OSA screening among professional male drivers. The small sample size of the current study and the inclusion of only male subjects as well as a high likelihood of response bias might play important roles in the current results. Further studies including a larger population and an objective test of fitness to drive are needed in this context.

## Figures and Tables

**Figure 1 jcm-13-05976-f001:**
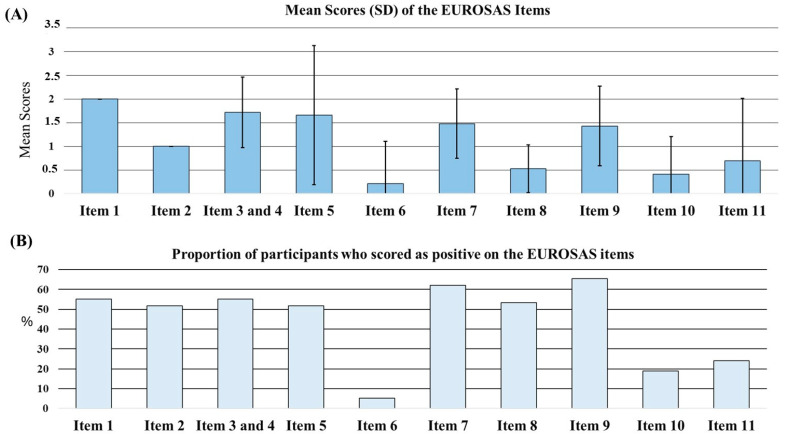
(**A**) The mean scores of each EUROSAS item across the entire study population. (**B**) The proportion of participants who were scored as positive on the items of EUROSAS. EUROSAS: European Obstructive Sleep Apnea Screening. SD: standard deviation.

**Figure 2 jcm-13-05976-f002:**
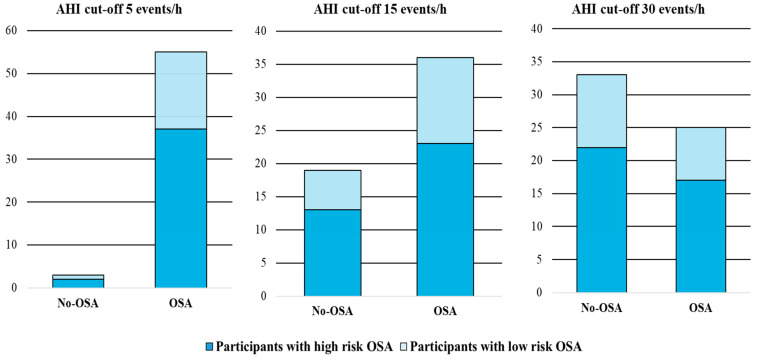
The number of the participants based on the EUROSAS vs. PSG results. AHI: Apnea Hypopnea Index. OSA: Obstructive Sleep Apnea. No-OSA: No Obstructive Sleep Apnea.

**Figure 3 jcm-13-05976-f003:**
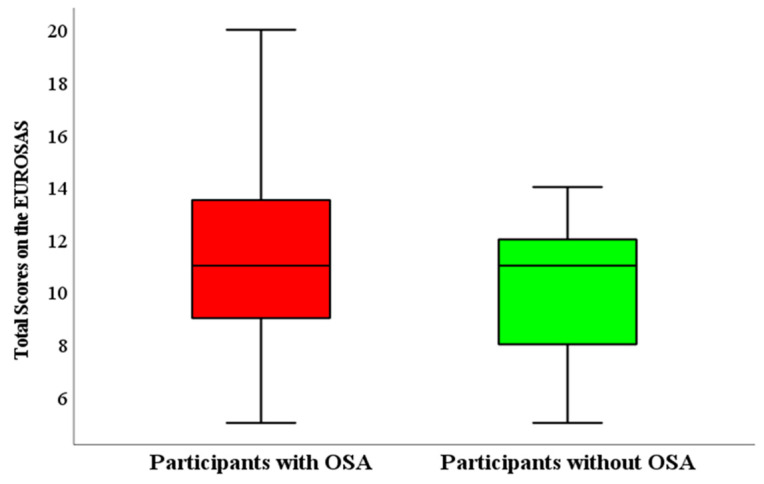
Comparison of the total EUROSAS scores across the patients with and without OSA. EUROSAS: European Obstructive Sleep Apnea Screening. OSA: Obstructive Sleep Apnea.

**Figure 4 jcm-13-05976-f004:**
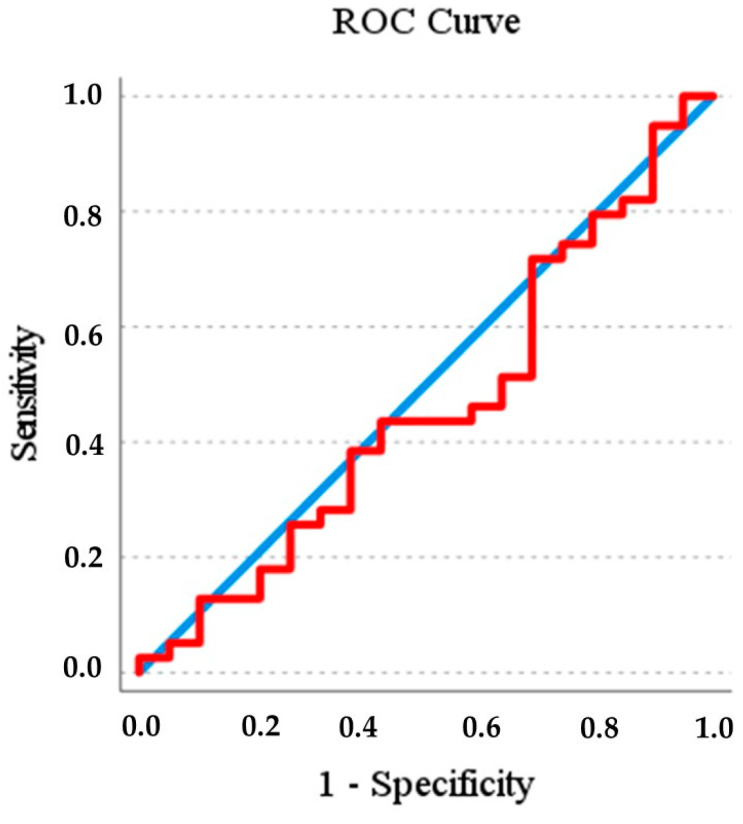
Receiver operating characteristic curve of association between the AHI measured by polysomnography and the EUROSAS result (low or high risk of OSA). ROC: Receiver Operating Characteristic.

**Table 1 jcm-13-05976-t001:** Baseline characteristics and demographics of the study population.

	Study Population (*n* = 58)	AHI ≥ 15 Events/h (*n* = 39)	AHI < 15 Events/h (*n* = 19)	*p*
** *Demographic Characteristics* **				
Age, years	45.7 (42.1–52.2)	45.9 (42.9–52.7)	44.9 (37.7–52.0)	0.451
BMI, kg/m	31.9 (28.6–34.4)	33.0 (29.4–34.9)	29.1 (26.7–32.3)	0.021
Obesity, *n* (%)				
Class I	47 (81)	31(79.5)	16 (84.2)	0.603
Class II	9 (15.5)	6 (15.4)	3 (15.8)	
Class III	2 (3.4)	2 (5.1)	0 (0)	
ESS	6.5 (3.0–10.3)	6 (3–11)	7 (3–9)	0.727
Sleepy, *n* (%)	14 (24.1)	11 (28.2)	3 (15.8)	0.350
Married	50 (86.2)	33 (84.6)	17(89.5)	0.615
High School Education	45 (84.9)	31 (86.1)	14 (82.3)	0.057
Current smoking	34(58.6)	21 (53.8)	13 (68.4)	0.628
Alcohol (every day /once a week)	7 (12.3)	4 (10.5)	3 (15.8)	0.238
**Vital Signs**				
SBP	120 (110–130)	123 (116–130)	120 (110–141)	0.473
DBP	80 (72–90)	80 (73–88)	80 (70–90)	0.819
Heart Rate	80.5 (73.0–95.0)	83 (76–96)	78 (69–91)	0.144
** *Comorbidities* **				
Asthma/COPD, *n* (%)	3 (5.3)	2 (5.3)	1 (5.3)	1.000
Hypertension, *n* (%)	10 (17.5)	8 (21.1)	2 (10.5)	0.325
Hyperlipidemia, *n* (%)	6 (10.3)	4 (10.3)	2 (10.5)	0.975
Diabetes Mellitus, *n* (%)	5 (8.6)	3 (7.7)	2 (10.5)	0.718
Stroke, *n* (%)	1(1.7)	1 (2.6)	0 (0.0)	0.481
Angina pectoris, *n* (%)	4 (6.9)	2 (5.1)	2 (10.5)	0.446
Cardiac failure, *n* (%)	1 (1.7)	0 (0.0)	1 (5.3)	0.148
Arrythmia, *n* (%)	5 (8.6)	3 (7.7)	2 (10.5)	0.718
AMI, *n* (%)	3 (5.2)	2 (5.1)	1 (5.3)	0.983
PCI or CABG, *n* (%)	3 (5.2)	2 (5.1)	1 (5.3)	0.983
Hyperthyroidism, *n* (%)	3 (5.2)	2 (5.1)	1 (5.3)	0.983
Psychological Disorders, *n* (%)	1 (1.7)	0 (0.0)	1 (5.3)	0.148
Neurological Disorder, *n* (%)	3 (5.2)	1 (2.6)	2 (10.5)	0.199

AHI: Apnea Hypopnea Index. BMI: Body Mass Index. ESS: Epworth Sleepiness Scale. SBP: diastolic blood pressure. DBP: systolic blood pressure. PCI: percutaneous coronary intervention. CABG: Coronary artery bypass grafting.

**Table 2 jcm-13-05976-t002:** Comparisons of the PSG results across the groups with high and low risks.

	High-Risk OSA (*n* = 39)	Low-Risk OSA (*n* = 19)	*p*
Total Sleep Time (TST)	381.9 (50.2)	389.0 (42.2)	0.42
Sleep efficiency, % of TST	83.3 (7.6)	84.7 (7.8)	0.57
Sleep latency, min	27.0 (17.0–48.0)	25.0 (17.6–40.0)	0.80
N1 sleep	15.0 (11.0–15.00)	16.0 (13.0–22.0)	0.69
N2 sleep	214.7 (47.3)	227.5 (50.9)	0.32
Slow wave sleep, min	86.8 (37.2)	79.6 (30.6)	0.74
Slow wave sleep, % of TST	22.6 (8.7)	20.9 (8.6)	0.59
REM duration, min	64.0 (52.0–84.0)	64.0 (51.0–78.0)	0.77
REM, % of TST	17.9 (6.4)	16.9 (6.4)	0.49
REM latency, min	129.0 (98.0–174.0)	140.2 (105.8- 266.8)	0.34
AHI, events/h	17.9 (11.0–43.0)	25.4 (14.4–45.5)	0.45
AHI REM, events/h	34.1 (21.0)	36.9 (26.5)	0.82
AHI, non-REM, events/h	20.2 (9.0–43.0)	23.1 (11.4–47.6)	0.54
AHI supine, events/h	50.1 (32.6)	49.1 (28.2)	0.99
ODI, events/h	15.5 (6.2–30.3)	21.7 (8.4–31.9)	0.70
Average SpO_2_, %	93.0 (91.7–93.7)	93.5 (91.3–94.0)	0.47
Minimum SpO_2_, %	82.0 (77.0–86.0)	80.0 (74.0–84.0)	0.46
SpO_2_ < 90%, min	82 (77–86)	80 (74–84)	0.46
Heart rate, bpm	67.3 (9.7)	66.4 (9.7)	0.86

OSA: Obstructive Sleep Apnea. N1: Non-REM stage 1. N2: Non-REM stage 2. REM: Repeat Eye Movement. ODI: oxygen desaturation index.

**Table 3 jcm-13-05976-t003:** Predictive parameters for the EUROSAS to identify AHI ≥ 5/15/30 events/h.

	AHI ≥ 5	AHI ≥ 15	AHI ≥ 30
DOR	1.02	0.92	1.06
Sensitivity	67.27 (53.29–79.32)	66.67 (49.78–80.91)	68.0 (46.50–85.05)
Specificity	33.3 (0.84–90.57)	31.58 (12.6–56.6)	33.33 (17.96–51.83)
PLR	1.0 (0.44–2.29)	0.97 (0.67–1.42)	1.02 (0.71–1.46)
NLR	0.98 (0.19–5.08)	1.06 (0.48–2.34)	0.96 (0.45–2.03)
Disease Prevalence	94.83 (85.62–98.92)	67.24 (53.66–78.99)	43.10 (30.16–56.77)
PPV	94.87 (89.06–97.68)	66.67 (57.82–74.47)	43.59 (35.00–52.58)
NPV	5.26 (1.06–22.34)	31.58 (17.22–50.59)	57.89 (39.43–74.39)
Accuracy	65.52 (51.88–77.51)	55.17 (41.54–68.26)	48.28 (34.95–61.78)

AHI: Apnea Hypopnea Index. DOR: diagnostic odds ratio. PLR, positive likelihood ratio. NLR: negative likelihood ratio. PPV, positive predictive value. NPV, negative predictive value.

## Data Availability

Data collected for the study, including de-identified individual participant data, will be made available to others within 6 months of the publication of this article, as will additional related documents (study protocol, statistical analysis plan, and informed consent form), for academic purposes (e.g., meta-analyses), upon request to the corresponding author (yupeker@ku.edu.tr) and with a signed data access agreement.

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
