# Peer review of "Validation of the European Obstructive Sleep Apnea Screening (EUROSAS) in Professional Male Drivers"

_jcm, 2024, doi:10.3390/jcm13195976_

Round 1

Reviewer 1 Report

Comments and Suggestions for Authors

It is interesting

I suggest to cite in the introduction some tests, questionnaires currently used in suspicious of OSA, such as STOP-Bang. 

In the methods :

please report the AHI value along with the average oxygen saturation

moreover information regarding the equipment used should be added.

Had the criteria regarding alcohol consumption been included?

I suggest in the methods to specify cutoffs that identify levels of obesity severity. Clarify whether the latest guidelines related to OSA have been used.

On line 121 please clarify what the comparison groups were. Why did you use those threshold values to evaluate the predictive parameters?

Since the high percentage of smokers in the study population ,I suggest to add a reference for discussion about the association of nocturnal breathing disorders and smoking

Clin Respir J. 2020 Jan;14(1):29-34. 

Reviewer 2 Report

Comments and Suggestions for Authors

The topic is interesting and the paper is quite well written. I have some comments:

1) Abstract. Results: In all, 39 (67.2%) cases were categorized as having high-risk OSA, while 19 (32.8%) were classified as low-risk OSA based on the EUROSAS. The PSG results revealed that 37 patients (94.9%) in the high-risk group and 18 patients (94.7%) in the low-risk group had OSA. The EUROSAS had a sensitivity of 67.2%, a specificity of 33.3%, a positive predictive value of 94.8%, a negative predictive value of 5.2%. Please, underline the most important statistically significant values to support the data. 

2) Abstract. Conclusions: The EUROSAS provides a moderate level of accuracy for the screening of OSA in the professional male drivers. Further research is needed for objective evaluation of fitness to drive tests in drivers with OSA. Abstract might be beneficial to include a sentence that briefly summarizes the key findings of the study. This can provide readers with a quick overview of the research. 

3) 1. Introduction 33 Motor vehicle accidents (MVAs) are one of the most important causes of death and 34 injury worldwide [1]. One of the significant risk factors contributing to MVAs is obstruc- 35 tive sleep apnea (OSA) which is a health-related condition characterized by repetitive ep- 36 isodes of airflow cessation due to upper airway collapse during sleep[1]. The estimated 37 prevalence of OSA ranges from 9% to 38%, while 80-90% of those individuals with OSA 38 is frequently undiagnosed in the general population [2, 3]. Given the consequences of 39 OSA such as permanent impairment of cognitive functions including attention, executive 40 functions and psychomotor speed and coordination, it is not surprising that the risk of 41 MVAs in OSA population is greater than that for the general population. 42 OSA appears to be much more prevalent in professional motor vehicle drivers due 43 to unique population characteristics such as higher proportion of males, obesity rates, and 44 age distribution [4]. The incidence rate of OSA has been reported between 28% and 78% 45 in commercial drivers [5]. A large number of studies have reported that commercial ....  Authors are kindly requested to emphasize the current concepts about these issues in the context of recent knowledge and the available literature. I think that these articles should be quoted in the References list. References: 1. Association of Low Arousal Threshold Obstructive Sleep Apnea Manifestations with Body Fat and Water Distribution. Life (Basel). 2023;13(5):1218. Published 2023 May 19. doi:10.3390/life13051218 

2. Low arousal threshold: a common pathophysiological trait in patients with obstructive sleep apnea syndrome and asthma. Sleep Breath. 2023;27(3):933-941. doi:10.1007/s11325-022-02665-4

3. A New Screening Tool (BAMSA) for Sleep Apnea in Male Professional Truck Drivers. J. Clin. Med. 202413, 522. https://doi.org/10.3390/jcm13020522

4) In this context, the European Union Driver License Committee developed a ques- 62 tionnaire as a screening tool for OSA, especially for drivers of motor vehicles, (Appendix 63 A)[15]. Previously, we named the questionnaire, the European Obstructive Sleep Apnea 64 Screening as “EUROSAS”, and reported that its test-retest reliability was poor among 65 male and female drivers [16]. Further validation study was needed due to lack of objective 66 sleep testing [16]. Therefore, the aim of the current study is to validate the EUROSAS 67 against the PSG in professional male drivers. Please, improve the decription of study aim.

5) 3. Results 133 3.1. Baseline Characteristics of the Study Population 134 Baseline characteristics of the study population as well as the study groups has been 135 presented in the Table 1. A total of 58 male professional drivers with a mean age of 46.9 136 (7.5) were included to the present study. Median ESS score was 6.5 (IQR 3.0-10.3), and 137 24.1% of the cohort were classified as sleepy (ESS score ≥11). Majority of the study popu‑ 138 lation were obese, married and graduated from high school. The participants with OSA 139 were more obese than the ones without OSA while other demographic characteristics and 140 comorbidities were similar. Please, underline in the manuscript the most important data to support the results.

6) 4. Discussion 196 The main finding of the current study is that the EUROSAS has a moderate predictive 197 ability to detect OSA in professional male drivers. The sensitivity and specificity rates 198 were similar against the different AHI thresholds, whereas the best performance of the 199 positive and negative predictive values was obtained using the AHI cut-off 5 events/h. The discussion section needs to be improved.  It could be interesting to clarify the results obtained and compare them with previous published literature.

7) 5. Conclusions 300 The EUROSAS questionnaire has a moderate level of accuracy for OSA screening 301 among professional male drivers. Objective OSA measures and fitness to drive tests are 302 needed in this context. I suggest to underline also in the conclusions the limitations of the study, because they significantly influence the conclusions. I also think it may be useful to underline that further studies with a larger population need to be carried out.

Comments on the Quality of English Language

Minor changes of English language are required.

Round 2

Reviewer 2 Report

Comments and Suggestions for Authors

The manuscript has been improved, as requested. No furher comments.